# Line Structure Extraction from LiDAR Point Cloud Based on the Persistence of Tensor Feature

**Xuan Wang [1], Haiyang Lyu [2], Weiji He [1,*] and Qian Chen [1]**

[1] Jiangsu Key Laboratory of Spectral Imaging & Intelligence Sense (SIIS), Nanjing University of Science and Technology, Nanjing 210094, China

[2] Smart Health Big Data Analysis and Location Services Engineering Research Center of Jiangsu Province, Nanjing University of Posts and Telecommunications, Nanjing 210023, China

\* Correspondence: hewj@njust.edu.cn; Tel.: +86-025-85866638

**Featured Application: line structure extraction, tensor voting, persistent homology.**

**Abstract:** The LiDAR point cloud has been widely used in scenarios of automatic driving, object recognition, structure reconstruction, etc., while it remains a challenging problem in line structure extraction, due to the noise and accuracy, especially in data acquired by consumer electronic devices. To address the issue, a line structure extraction method based on the persistence of tensor feature is proposed, and subsequently applied to the data acquired by an iPhone-based LiDAR sensor. The tensor of each point is encoded, voted, and aggregated by its neighborhood, and further decomposed into different geometric features in each dimension. Then, the line feature in the point cloud is represented and computed using the persistence of the tensor feature. Finally, the line structure is extracted based on the persistent homology according to the discrete Morse theory. With the LiDAR point cloud collected by the iPhone 12 Pro MAX, experiments are conducted, line structures are extracted from two different datasets, and results perform well in comparison with other related results.

**Keywords:** line structure extraction; tensor feature decomposition; persistent homology; iPhone-based LiDAR sensor

## 1. Introduction

Light detection and ranging (LiDAR) is a method to measure the distance between the object and the receiver based on the reflection of light and obtain massive points for the surface of an area instantly [1–3]. With high efficiency and versatile performance, it has been extensively applied in surveying and mapping, automatic driving, scene understanding [4,5], etc. In recent years, LiDAR sensors have been equipped in consumer electronic devices, such as Kinect, iPhone, etc., which has made it more and more convenient to acquire point cloud data for common users, and many applications have been constructed [6,7], such as line structure extraction, shape recognition, object detection and classification, and some high-level applications (scene understanding, simultaneous localization and mapping, etc.).

With these easy-to-use LiDAR devices, plenty of point cloud datasets are provided, and abundant information can be extracted, such as geometric feature, semantic labeling, and scenario relations [8]. However, it also brings many problems in dealing with these redundant point cloud datasets, since not all points are needed, and it's difficult to extract the structure information, compress the data, and represent these redundant datasets by simple geometric structures [9,10]. To solve these problems and obtain structure information, different studies have been conducted using, e.g., deep learning-based feature extraction methods and geometric model fitting methods [7,8,11,12]. These methods either need predefined geometric models [12,13] or large amounts of training datasets [7]. In addition, some post-processing operations are also needed to maintain connection relations of geometric structures, and results can be affected by the quality of the point cloud datasets [14,15].

(1) How to extract line information from the quality-unstable point cloud dataset collected by consumer electronic devices, and (2) how to construct line structures without predefined geometric models and manually selected training datasets, remain challenges.

To address the issue, a line structure extraction method based on the tensor voting [16,17] and the persistent homology theory [18–20] is proposed. The line structure extraction framework is designed for point cloud datasets collected by the iPhone-based LiDAR sensor. The line feature in each point is encoded by the tensor voting, then the line segment is represented as the connection of each point with the highest local line feature value, and the line structure is constructed from line segments along with their structure connections. Contributions are as follows: (1) We compute the line feature from the point cloud based on the tensor voting theory, decompose the tensor feature, and make a combination of different dimensional geometric features, to get the line candidate dataset. (2) We represent line feature by the tensor of the Morse–Smale complex, calculate the line structure from LiDAR point cloud based on the persistent homology theory, and extract the line structure using the connect relations of critical points. (3) We propose the unified framework, design the algorithm to extract line structure from point cloud collected by the iPhone-based LiDAR sensor, and make a comparison between line structure extraction results.

The remainder of this paper is structured as follows. The next section gives an overview of works related to this paper. Section 3 provides a detailed description of the line structure extraction method based on the persistence of tensor feature, including tensor voting, tensor feature decomposition, critical point representation in the discrete Morse theory, and the line structure extraction. Experiments are conducted. The results are discussed in Section 4, followed by conclusions in Section 5.

## 2. Related Works

Line structure is the concise representation of 3D scenes, and the process relates to description of line features, segmentation and recognition of objects, and scene understanding of point cloud datasets [5–8], etc.

One of the heated research fields is edge map detection from depth or RBG image, to get the layout information of indoor scenes [2,8]. The edge feature is computed by view relations between the 3D scene to the 2D image, using neural network or some image-based edge detection algorithms. Hence, it's also applied to the 3D point cloud [11,21,22]. Chen et al. [21] proposed a 3D line segment detection method from the multi-view stereo: project 3D points into planar sets on different camera matrices and computing line relations based on 2D images, then remove false matching relations, cast line features back to the 3D space of point clouds, and get 3D line segments. Another similar line segment detection strategy is the work conducted by Lu et al. [22]. The point cloud dataset is segmented into super planes based on region growing and merging. Then, points belonging to each plane are projected onto it, and 3D points in each plane become 2D images, followed by line extraction using 2D image contour extraction and least-square-fitting based line detection. In these line feature extraction methods, 3D line features can be duplicated or lost during dimensional transformation. However, line structure extraction directly from 3D point cloud can be conducted by an intuitive way. The first kind of intuitive method is the 3D shape detection from 3D point cloud, according to fitting error between the point cloud and the predefined geometric model. Hough transformation [13] is usually applied to shape detection situations, especially line segments, along with the random sample consensus (RANSAC) [12], which can extract geometric shapes from noisy point cloud data; besides, there are also some shape detection methods based on neural network that also works for the similar propose. Nevertheless, these methods depend on the predefined geometric model, which limited the capability of application. The other kind of intuitive method is the line structure extraction based on the 3D point cloud segmentation [1,4,23,24], where the point cloud dataset is directly divided into different object groups, and the rim of geometric objects is extracted to construct edge structures. Gankhuyag et al. [24] proposed an automatic 2D floorplan CAD generation method based on plane detection,

and points segmented into floors are preserved to build 2D CAD line segments. For 3D line structure detection, Lin et al. [4] proposed a line extraction method by segmenting point cloud into facets and extracting boundary of each facet to get the 3D line. Although the structure information can be further refined using some semantic information, lines are extracted under the assumption of predefined geometric model or rules, i.e., the straight-line restriction. There are also some neural-network-based point cloud segmentation methods [5,7,11] that can extract boundaries more precisely than the predefined geometric models or rules, but these methods need large amounts of training data. In addition, some post-processing operations are also applied to maintain connection relations of geometric structures.

Another heated research field is line structure extraction from point clouds based on the line feature representation [25,26]. Using the consecutive geometric feature contained in point cloud, line structures can be detected and traced along the consecutive direction of geometric feature [6,9,10,27,28], such as angle differences between the normal vector and its neighboring points, curvature computed from assumed point cloud surface, etc. These methods usually need normal vectors recorded in the point cloud dataset. While there is no normal information stored, some techniques need to be applied to compute normal vectors [6,9,27], such as local plane fitting, principal component analysis (referred to as PCA), and tensor voting. In PCA-based methods, the normal vector of point is computed using the local covariance matrix based on its neighborhood [27], and different dimensional geometric features can be interpreted from eigen decomposition of the matrix; besides, the optimal size of neighborhood is the critical point and is affected by the quality of point cloud dataset [10]. In tensor-voting-based methods, it can deal with the noisy dataset using the tensor encoding and revoting process and has the capability of N-dimensional geometric feature representation [6,15,16,28,29]. Using tensor voting, normal vector of the point is computed from its neighborhood and subsequently revoted to get the refined result. Then, geometric features of different dimensional are provided, and the subset of the point cloud dataset with the label of line feature is extracted [29]. However, the next step of extracting line structure from the selected line subset, still remains to be a challenge. Compared with geometric-model-based shape detection methods, the topological relation-based method can deal with the graph connection of line structures. One of the widely used topological methods is the persistent homology [30–32], where topological relations are computed and captured based on N-dimensional holes during growth of the distance to merge all points. Using the persistent homology theory, Carlsson [19] computed topological patterns contained in a point cloud dataset and built pattern signatures by barcodes. Another form of topological features computed based on the persistent homology is studied by Zhou et al. [20] and Wong et al. [33], and topological features are presented by the persistent image, which is further computed using the persistent diagram. Moreover, Beksi et al. [34] presented a 3D point descriptor according to birth and death of 0-cycles and 1-cycles and took it as topological signatures to classify categories for noisy point cloud datasets. These topological methods can deal with quality-unstable point cloud dataset using the topological feature and preserve graph connections with each line segment, while it is difficult to maintain geometric shapes in the dataset.

To deal with the quality-unstable point cloud dataset and obtain connection relation preserved line structures, a line structure extraction method is proposed in this paper, and point cloud datasets are collected using the iPhone-based LiDAR sensor. In addition, structure refinement based on the assumed geometric rules is not going to be conducted in this paper, since line structures are extracted with no predefined geometric models and no manually selected training datasets.

### 3. Methodology

Suppose a point $p$ is consisted of $N$-dimensional coordinate and without normal vector information, which can be denoted as $p(x_1, x_2, \ldots, x_N)$. The line feature is the geometric information of each point, and the higher value of the line feature, the more probability of

the point to be a part of lines. Hence, the line segment can be constructed by connecting each point with the highest local line feature value. The line structure is built using line segments and their structure connections. To get the line structure from the unstructured point cloud, the geometric feature to express line feature needs to be constructed, which is computed based on the tensor voting, then line datasets to extract line segments is generated. The next step is extracting consecutive line segment from the subset, which is conducted based on the persistence of tensor feature, using topological relations calculated by the persistent homology. Hence, the line structure extraction process consists of 4 steps:

(1)　initialize the tensor for unstructured point and get geometric features in different dimensions;

(2)　revote the tensor feature in different dimensions and compute the refined geometric feature;

(3)　represent the geometric feature and construct the line candidate subset (referred to as *LCS*);

(4)　construct the Morse–Smale complex and extract line structures based on the discrete Morse theory.

The pipeline of the line structure extraction framework is depicted in Figure 1.

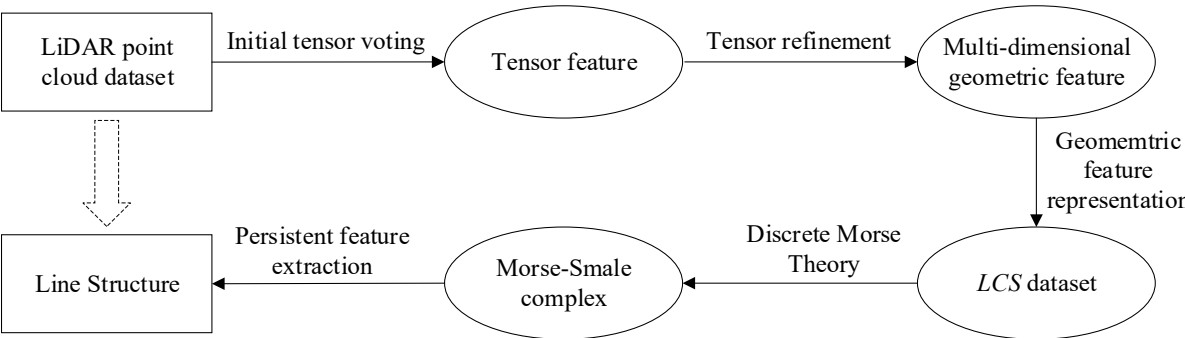

**Figure 1.** The pipeline of the line structure extraction framework.

### *3.1. Initial Tensor Voting and Dimensional Feature Presentation*

According to the manifold theory, the *N*-dimensional space *S* is composed of *d*-dimensional normal subspace $S_N$ and $(N-d)$-dimensional tangent subspace $S_{N-d}$, which are orthogonal and complementary to each other, as denoted in Equation (1). The geometric feature in each dimension can be detected based on the normal subspace $S_d$, where the normal vector depicts the orthogonal direction of geometric structure. For example, in the 3-dimensional space (*N* = 3), the geometric structure with 1-dimensional normal space $S_1$ can be detected as "surface," the geometric structure with 2-dimensional normal space $S_2$ can be detected as "line", and the geometric structure with 3-dimensional normal space $S_3$ can be detected as "point".

$$S = S_d + S_{N-d} \; \& \; S_d \cdot S_{N-d} = 0 \tag{1}$$

However, if there is no normal recorded, the normal space can be computed using tangent information from its neighborhood. Suppose a point *p* with a point $p_i$ fromneighborhood Ω, and the vector from $p_i$ to *p* can be denoted as $v_i$, which contains the tangent information of $p_i$. Then $v_i$ is normalized as $\hat{v}_i$ and the tangent subspace $S_{N-d}^i$ can be computed based on the Kronecker delta of $\hat{v}_i$, as denoted in Equation (2).

$$S_{N-d}^i = kron\left(\hat{v}_i, \hat{v}_i^T\right) \tag{2}$$

Since there is no normal vector recorded and no preferred geometric features set for space *S*, it can be denoted as the identity matrix *I* for space *S*. Hence, the geometric structure encoded by normal subspace $S_d^i$ that will be transmitted from point $p_i$ to *p* can be denoted

as complementary form of $S_{N-d}^i$ according to Equation (1), i.e., $S_d^i = I - S_{N-d}^i$. As seen in Figure 2, the geometric feature of each point in neighborhood of $p$ is voted to the point, with geometric information transmitted by the normal subspace.

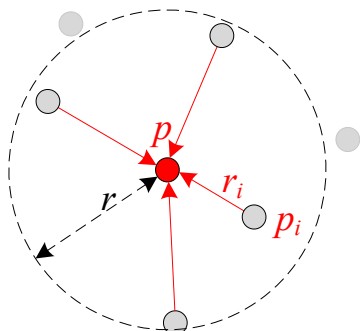

**Figure 2.** The tensor voting process of the point with its neighborhood.

Besides, the intensity of geometric feature decays along distance $r_i$ from $p_i$ to $p$, and the decay function is usually taken as the Gaussian function, i.e., $w(r) = e^{-\left(\frac{r}{\rho}\right)^2}$. Here, the $\rho$ for $w$ is set to be 0.1. Finally, tensor $T_p$ at point $p$ is computed and accumulated using that of the point from its neighborhood, as denoted in Equation (3).

$$T_p = \sum_{p_i \in \Omega} w(r_i)\left(I - S_{N-d}^i\right) \tag{3}$$

### 3.2. The Refinement of Initial Tensor Using Different Dimensional Geometric Feature

With tensor $T$ computed through initial tensor voting process, the geometric feature in each dimension can be computed based on the tensor decomposition of $T$. In $N$-dimensional space, $T$ is presented by a $N \times N$ dimensional matrix, and can be further decomposed by eigenvectors $\left(\vec{e}_1, \vec{e}_2, \cdots, \vec{e}_N\right)$ and related eigenvalues $(\lambda_1, \lambda_2, \cdots, \lambda_N)$, where $\lambda_1 \geq \lambda_2 \geq \cdots \geq \lambda_N$. Suppose the $d$ dimensional geometric feature is going to be transmitted from point $p(x_1, x_2, \ldots, x_N)$ to $q(x_1, x_2, \ldots, x_N)$, the normal vector of normal subspace is denoted as $v_n = \left(\vec{e}_1, \vec{e}_2, \cdots, \vec{e}_d\right)$, the tangent direction vector is denoted as $v_t = \left(\vec{e}_{d+1}, \vec{e}_{d+2}, \cdots, \vec{e}_N\right)$, and vector from $p$ to $q$ is denoetd as $v = q - p$. Then, the normal vector $v_k$, which is transmitted along a minimal circle path defined by points $p$ and $q$, meets the relations defined in Equation (4).

$$v_k = v_n \cos(2\theta) - v_t \sin(2\theta) \tag{4}$$

In Equation (4), angle $\theta$ is computed based on cosine value and dot multiply operation using geometric relations of $v_t$, and $v_k$, i.e., $\theta = a\cos(dot(norm(v_k), v_t)$. Based on Equations (2)–(4), the $d$ dimensional tensor feature $T_d$ transmitted from $p$ to $q$ can be encoded as Equation (5).

$$T_d = w(r, \theta)kron\left(v_k, v_k^T\right) \tag{5}$$

$w(r, \theta) = e^{-\left(\frac{r}{\rho}\right)^2}(\cos(\theta))^4$ is the decay function with distance $r$ and curvature information encoded as $\theta$. It has been proven that the angle $\theta$ will become 0 where dimension $d \geq 2$, and there is no need to recompute $\theta$ in each dimension, if $v_k$ is projected to $S_d$ (King [16]). Hence $d$-dimension tensor feature $T_d$ can be represented as Equation (6).

$$T_d = \begin{cases} w(r, \theta)kron\left(v_k, v_k^T\right) & d = 1 \\ w(r)\left(S_d - kron\left(v_n, v_n^T\right)\right) & d \geq 2 \end{cases} \tag{6}$$

On the other hand, the intensity $s_d$ of $d$ dimensional geometric feature can be represented using eigenvalues $(\lambda_1, \lambda_2, \cdots, \lambda_N)$, as denoted in Equation (7).

$$s_d = \begin{cases} \lambda_d - \lambda_{d+1} & d < N \\ \lambda_N & d = N \end{cases} \tag{7}$$

Based on Equations (6) and (7), tensor voted from $p$ to $q$ in each dimension can be computed by Equation (8).

$$T^p = \sum_{d=1}^{d \leq N} s_d T_d \tag{8}$$

Finally, the tensor $T^q$ of point $q$ is voted by tensor $T^p$ of each point $p$ in p's neighborhood $\Omega$, as denoted in Equation (9), and this process is called the tensor refinement. The reader can refer to Wang, et al. [6] for a detailed derivation of the tensor refinement process.

$$T^q = \sum_{p \in \Omega} T^p \tag{9}$$

### 3.3. The Construction of LCS Based on Geometric Saliency

In 3D space, the point is composed of ($x$, $y$, $z$), which is just position information and no normal vector information recorded, and refined voting results is decomposed by eigenvectors $\left( \vec{e}_1, \vec{e}_2, \vec{e}_3 \right)$ and related eigenvalues $(\lambda_1, \lambda_2, \lambda_3)$, where $\lambda_1 \geq \lambda_2 \geq \lambda_3$. The sailency of geometric feature that encoded in each dimension can be computed based on Equation (7):

(1)  in 1-dimensional normal space $S_1 = kron\left( \vec{e}_1, \vec{e}_1^T \right)$, the geometric feature turns out to be "surface", since there is only the 1D stick-shaped normal vector, and geometric saliency is $s_{surface} = \lambda_1 - \lambda_2$;

(2)  in 2-dimensional normal space $S_2 = kron\left( \vec{e}_1, \vec{e}_1^T \right) + kron\left( \vec{e}_2, \vec{e}_2^T \right)$, the geometric feature turns out to be "line", since there is the 2D surface-shaped normal vector, and geometric saliency is $s_{line} = \lambda_2 - \lambda_3$;

(3)  in 3-dimensional normal space $S_3 = kron\left( \vec{e}_1, \vec{e}_1^T \right) + kron\left( \vec{e}_2, \vec{e}_2^T \right) + kron\left( \vec{e}_3, \vec{e}_3^T \right)$, the geometric feature turns out to be "point", since there is the 3D ball-shaped normal vector, and geometric saliency is $s_{point} = \lambda_3$.

In above descriptions, the higher geometric saliency value of the point, the more credible the geometric feature category it belongs to; for example, if the point with line geometric saliency $s_{line}$ higher than the saliencies of other dimensions, it can categorize into *LCS*. However, if the point with surface saliency $s_{surface}$ lower than some threshold value, it can also be taken into the *LCS*. Besides, the point with high point saliency $s_{point}$ turns out to be the corner structure of geometric framework, hence, it can also be taken into the *LCS*. With these assumptions, the *LCS* can be constructed based on threshold values $\left\{ \sigma_{surface}, \sigma_{line}, \sigma_{point} \right\}$ for the saliency in different dimensions, as dented in Equation (10).

$$LCS = \left\{ p\left( x, y, z, s_{surface}, s_{line}, s_{point} \right) \in S \middle| s_{surface} \leq \sigma_{surface} \cup s_{line} \geq \sigma_{line} \cup s_{point} \geq \sigma_{point} \right\} \tag{10}$$

### 3.4. Line Structure Extraction Based on the Discrete Morse Theory

Suppose $f : S(p) \to R$ is a smooth function on a manifold, and $\{p, f(p)|p \in S\}$ can be taken as a smooth surface in 3D space. Besides, the gradient $\nabla f(p)$ of the function $f$ at point $p$ is the direction $f(p)$ decrease with the largest rate, and the integral line $\iota$ on the manifold is defined as the maximal path passing through $p$ whose vectors agree with gradient $\nabla f(p)$. The start and end point of the integral line are critical points, which are non-degenerate and with gradient values $\nabla f(p) = 0$. In other words, the critical point can be the maxima, minima, or saddle. Hence, the 1-stable manifold $\iota(p)$ of the critical point

$p$ can be expressed as the line structure whose integral line ends at $p$ (i.e., the maxima). Intuitively, the 1-stable manifold can be taken as the ridge of the surface, where each point of 1-stable manifold is the local maxima if there is a line profile which is orthogonal to the direction of gradient direction of the point, as depicted in Figure 3.

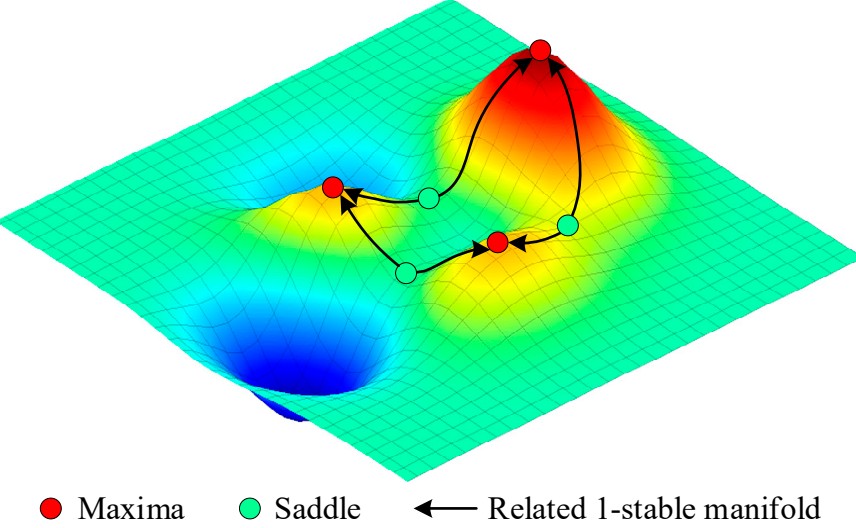

● Maxima ● Saddle ⟵ Related 1-stable manifold

**Figure 3.** 1-stable manifold of the critical point.

For the line structure extraction, line segments can be extracted with the local maximal line saliency value, i.e., the integral line starts from the saddle to the maxima (the black line in Figure 3), as denoted in Equation (11).

$$\iota(p) = \{p\} \cup \{q \in S_{LCS} | dest(q) = p\} \tag{11}$$

Using the discrete Morse theory, 1-stable manifold of the integral line can be encoded by the Morse–Smale complex, and the persistent geometric feature can be computed. Then 1-stable manifold $\iota(p)$ starting from the saddle to the maxima can be extracted. To construct the Morse–Smale complex from the quality-unstable point cloud dataset, the $LCS$ is further resampled by regular grid of space $S$, and the new $LCS'$ is computed as follows:

(1) compute the bounding box of space $S$, take the minimum edge as the referenced length, and divide it into $\kappa$ sub-edges of equal length $\tau$;

(2) divide the edge of bounding box in other 2 dimensions using the length $\tau$. Then, space $S$ is divided into subspace $\varsigma_i$ of equal size. Take the center position $center(\varsigma_i)$ of $\varsigma_i$ as the coordinate of the point in $LCS'$;

(3) compute the relation $\varphi_i : \left\{ p\left(x, y, z, s_{surface}, s_{line}, s_{point}\right) \middle| p \in LCS \right\} \to \varsigma_i$ between the point $p_i$ in $LCS$ and the subspace $\varsigma_i$, and take the point with the max line saliency as the new attribute for $\varsigma_i$, as denoted in Equation (12);

$$LCS' = \left\{ p\left(x, y, z, s'_{line}\right) \middle| (x, y, z) = center(\varsigma_i), s'_{line} = max(\{s_{line}\}) \right\} \tag{12}$$

(4) count the number $\delta_i$ of $\varphi_i$ in each $\varsigma_i$, and label $\varsigma_i$ with $\delta_i > 0$ as the mask area $\varsigma'_i$ of mask space $S' = \left\{ \varsigma'_i | \delta_i > 0 \right\}$, for the computation of persistent homology.

After the resample process, the Morse–Smale complex is constructed based on the discrete Morse theory, and the persistence of the geometric structure is computed. The line segment is extracted using the threshold $\delta$, and line features with higher persistent value than $\delta$ are preserved, i.e., $\{p_{critical}, l_{connection}\} = \left\{ \iota(p) \middle| \xi_{\iota(p)} > \delta \right\}$. Finally, the graph is computed and the line structure $\{\iota\}$ is constructed with connection information stored in the graph.

### 3.5. The Algorithm of the Line Structure Extraction Framework

The line structure extraction process is consisted of 4 steps. First, compute the initial tensor using the tangent space and decompose geometric features in different dimensions. Second, revote the tensor to get refined results and construct the *LCS*. Third, calculate the Morse–Smale complex for the *LCS* and get persistent features. Finally, extract the persistent line structure based on connection relations of critical points. The algorithm is depicted in Algorithm 1.

---

**Algorithm 1** The algorithm of the line structure extraction framework.

---

Line structure extraction framework **LSE**($P,r,\sigma,\kappa,\delta$)
**INPUT**: point cloud $P$, searching distance for neighborhood $r$, saliency thresholds
$\sigma\{\sigma_{surface}, \sigma_{line}, \sigma_{point}\}$, resampled grid $\kappa$, persistence threshold for line segment $\delta$
**OUTPUT**: line structure with connection relations $\{\iota\}$
*//step 1: compute the initial tensor and decompose geometric features in different dimensions*
**FOREACH** $p$ **in** $P$ *//for each point in P*
$\Omega\{p_i \in P | norm(p_i - p) \leq r\}$ = Neighborhood($p, r$);
$T^p$ = TensorVoting($p,\Omega$); *//compute initial tensor based on Equations (1)–(3)*
$\left\{\vec{e}, \lambda\right\}^p$ = GeoFeatureDec($T^p$); *//eigen decomposition for tensor $T^p$*
**END**
*//step 2: revote tensor and compute refined geometric feature*
**FOREACH** $p$ **in** $P$ *//for each point in P*
**FOREACH** $d$ **in** $N$ *//for each dimension in N, based on Equations (4)–(7)*
$s_d$ = SaliencyInDimD($\left\{\vec{e}, \lambda\right\}^p$); *//compute dimensional saliency*
$T_d$ = TensorInDimD($\left\{\vec{e}, \lambda\right\}^p$); *//compute dimensional tensor*
**END**
$T^p$ = AggTensor ($s_d, T_d$); *//aggregate tensor in each dimension based on Equation (8)*
$T$ = RevoteTensorFromNeig($T^{p \in \Omega}$); *//refine the voting result based on Equation (9)*
**END**
*//step 3: represent geometric feature and construct the LCS*
$s\left\{s_{surface}, s_{line}, s_{point}\right\}$ = DimSaliency($T$); *//compute dimensional saliency for each point*
$LCS$ = LineCandidateSubset($s,\sigma$); *//compute the LCS based on Equation (10)*
*//step 4: extract line structure based on the discrete Morse theory.*
$\{LCS', S'\}$ = ResampleLCS($P, \kappa, LCS$); *//resample the LCS based on Equation (11)*
$LinePer$ = MorseSmaleComplex($LCS', S'$); *//compute line structure based on Equation (12)*
$LineSeg\{p_{critical}, l_{connection}\}$ = LineExtract($LinePer, \delta$); *//extract line segment using threshold$\delta$*
$LineStr\{\iota\}$ = BuildLineStructure($LineSeg$); *//build line structure*
**RETURN** *Line Structure*$\{\iota\}$

---

## 4. Experiments and Discussions

This section focuses on the performance of line structure extraction method. The point cloud dataset acquired by the iPhone-based LiDAR sensor is collected, and tensor voting is conducted to get the *LCS*. Then, the line structure is extracted using the persistent line feature based on the discrete Morse theory. Comparisons are conducted and assessed with different methods. In addition, the line structure in a complicated scene is computed and evaluated.

### 4.1. The Dataset Collected by the iPhone-Based LiDAR Sensor

The point cloud dataset is acquired by the LiDAR sensor assembled in the iPhone 12 Pro MAX, which is a kind of consumer electronic device. It is a solid-state device based on the time of flight (referred to as TOF) technology, with a scanning range of 5 m and absolute accuracy of 0.01 m, as depicted in Figure 4. In addition, the mean square error in an indoor area of 1.49 m$^2$ is 0.0018 m [6]. The point cloud acquired by the LiDAR sensor is a scene of parterre, in an area of 7.76 m$^2$. To deal with the irregularly distributed sampling density, the dataset is randomly resampled with minimum distance of 0.01 m, and the number of

points in the dataset is 23,149, with the average number of 2400 points per m$^2$. As seen from Figure 4, the parterre is located on the plane and is consisted of four vertical sub-places and one flat rectangle ring plane, and the center plane inside the rectangle ring plane is a bare soil ground with uneven surfaces. Besides, there are noisy points and different point densities with unstable qualities in the randomly sampled point cloud data.

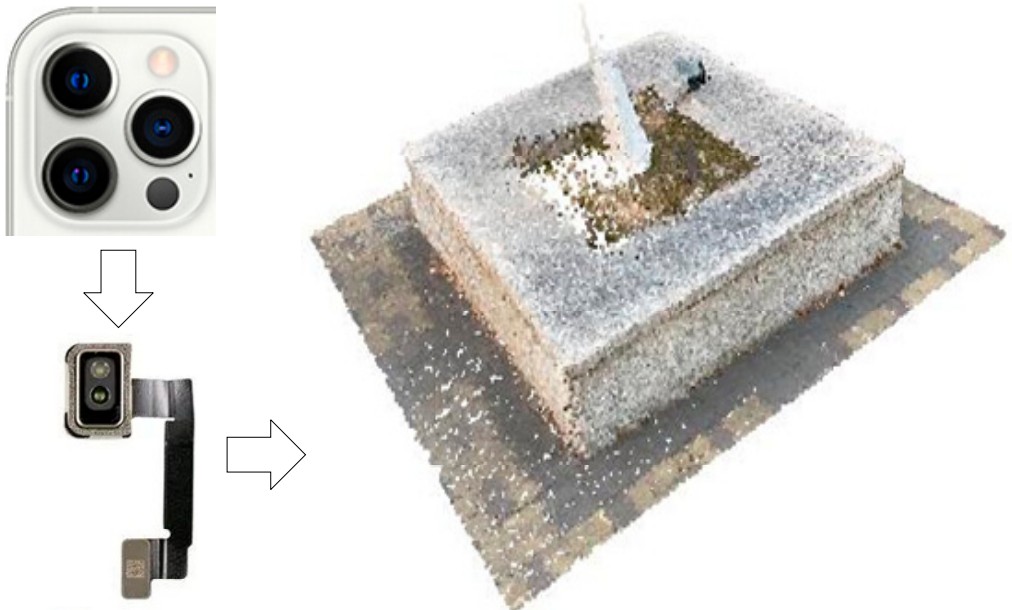

**Figure 4.** The iPhone-based LiDAR sensor and the collected point cloud dataset.

### 4.2. Tensor Voting and Geometric Dimension Representation

The point cloud $P$ is supposed to be the 3D point with only coordinate information as $(x, y, z)$. Hence, the initial tensor voting is conducted for each point $p$ with its neighborhood. Then, the normal vector in each dimension is computed for the space S, and the tensor in normal subspace and tangent subspace are represented. In addition, the searching distance for the neighborhood $r$ is set to be 0.3 m, and the minimum number of points involved in the tensor voting process is 5. In the second round of tensor voting, also called tensor refinement, the geometric feature in each dimension is encoded and revoted by its neighborhood. With the refined tensor, the geometric feature is recomputed, and different dimensional geometric feature is labeled with the saliency $s_{surface}, s_{line}, s_{point}$, which can be taken as the attribute of each point, i.e., $p\left(x, y, z, s_{surface}, s_{line}, s_{point}\right)$, depicted in Figure 5.

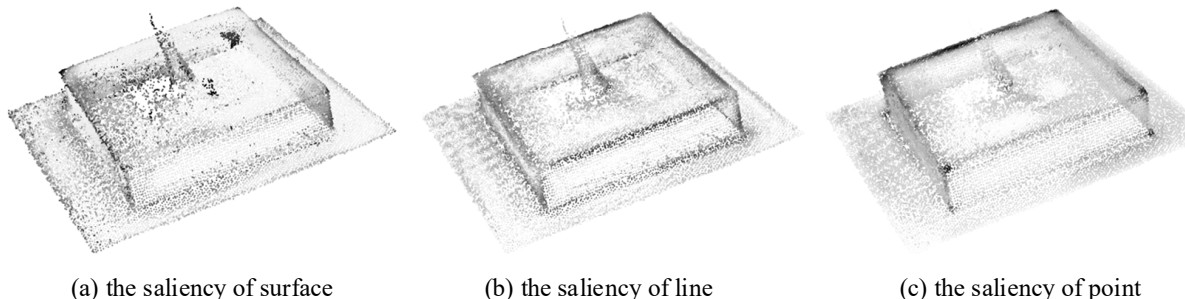

(a) the saliency of surface      (b) the saliency of line      (c) the saliency of point

**Figure 5.** The tensor voting results in different geometric dimensions.

As seen in Figure 5, saliencies of different dimensional geometric features encoded by the tensor are shown by the colormap ranging from light to dark. In Figure 5a, the surface feature of each point is labeled from black to white, and the lower the surface saliency, the

darker the color, and vice versa. However, in Figure 5b, the line saliency of each point is labeled in the opposite way, as well as the point saliency in Figure 5c.

### 4.3. The LCS Construction and Resampling

It can be obviously observed that the point with high line saliency in Figure 5a is considered to show a line structure, while the point with low surface saliency in Figure 5a also turns out to show a line structure. Besides, the point with high point saliency in Figure 5c seems to be the key point of show line structure, although some noisy points also have high point saliencies. Hence, the *LCS* can be constructed based on the threshold values $\left\{ \sigma_{surface}, \sigma_{line}, \sigma_{point} \right\}$. According to the saliency of the computation result, $\sigma_{surface}$ is set to be 3.27, and the point with saliency lower than the threshold is selected. However, $\sigma_{line}$ is set as 3.56 and $\sigma_{point}$ is set as 6.33, and the point with saliency higher than the threshold is selected. With the threshold of different geometric features, the *LCS* in different dimension is computed, and the statistic are depicted in Table 1.

**Table 1.** The statistic of *LCS* by different geometric features.

| Different Geometric Feature | The Number of Points |
|---|---|
| *LCS* by $s_{surface}$ | 86 |
| *LCS* by $s_{line}$ | 3326 |
| *LCS* by $s_{point}$ | 982 |
| *LCS* without duplication | 4155 |
| ground truth *LCS* | 4006 |

As seen from Table 1, the number of the *LCS* point computed using $s_{surface}$, $s_{line}$, and $s_{point}$ is 89, 3326, and 982, respectively, and the number of ground truth *LCS* computed using the spatial relation (within the distance of 0.05 m) between the ground truth line structure and the point cloud, is 4006. Only using single geometric feature, the *LCS* cannot cover the entire line structure. In addition, as depicted in the fourth row of Table 1, the number of the LCS without duplication is 4155, which is a little higher than that of the ground truth *LCS*. Hence, although there might be redundancy, it is reasonable to select the *LCS* based on Equation (10). Finally, the *LCS* is constructed, as shown in Figure 6.

To deal with unstable quality point cloud and unevenly distributed point density, the *LCS* is further resampled as the regular grid point of the space *S*. The more complex the scene, the higher the grid size $\kappa$ to preserve the detailed line structure, and vice versa. However, it is not a good idea to set $\kappa$ with a very high value since it may cause broken line segments in the result. For example, if grid size $\kappa$ is set to be too high such that gird interval is smaller than the minimum distance between two neighboring points, there will be vacant grids and the consecutive line segment will become broken. Hence, the suggested ranging domain is [10, 100]. With grid size $\kappa = 45$, the point in *LCS* is resampled and becomes a regularly distributed *LCS'* in the space *S*. Besides, the computation complexity for the line structure extraction is also simplified and refined.

### 4.4. Comparisons with Other Line Structure Extraction Methods

With the *LCS'* computed in Section 4.3, the Morse–Smale complex is constructed based on the discrete Morse theory, and the persistence of complex structures is computed. The important geometric structure will keep a high persistence value during the filtration of the Morse–Smale complex structure. Moreover, the persistence of 1-stable manifold is the curve structure for the *LCS'*. Hence, the higher the persistence, the more important the line segment. Finally, line segment $\left\{ p_{critical}, l_{connection} \right\}$ is extracted from the *LCS'* based on the persistence threshold $\delta$. Then, a graph is built and line structure $\{\iota\}$ is computed using the line segment and connection relations, as depicted in Figure 7.

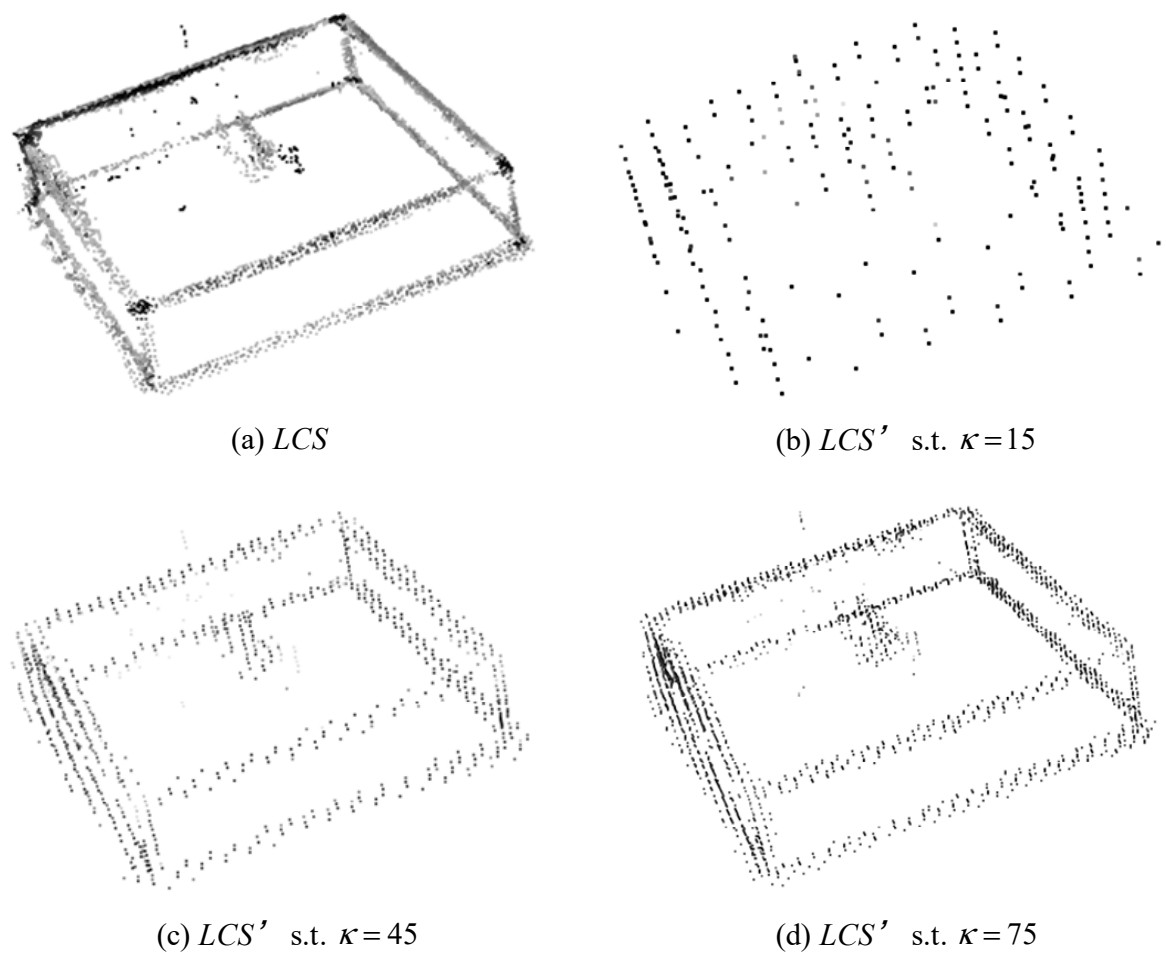

**Figure 6.** The *LCS* and *LCS′* with different t grid sizes *κ* of the point cloud.

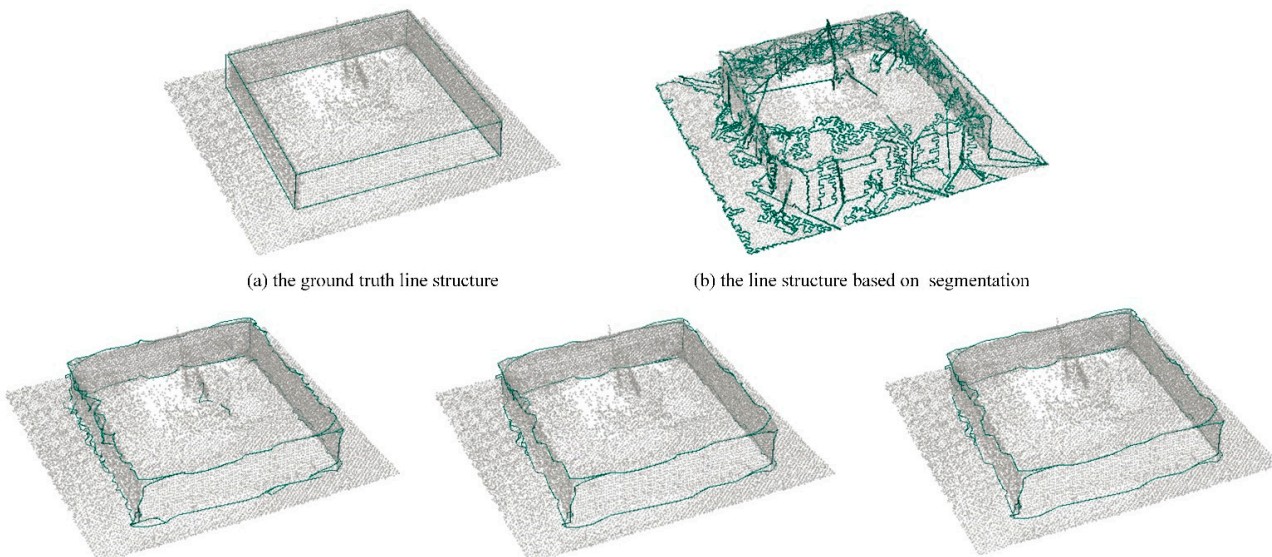

**Figure 7.** The comparison of different line structure extraction results.

As shown in Figure 7, the ground truth line structure is shown in Figure 7a, and it's manually drawn from the original dataset. Figure 7b shows the result constructed based on the point cloud segmentation method [23] (with parameter {Max angle: 20°, Max relative

distance: 1.0 m, Max distance: 0.2 m, Min points per facet: 10, Max edge length 0.03 m}), and the line structure is extracted using the rim of segmented geometric objects, while the result does not seem reasonable when compared with Figure 7a, since there are many mistakes in the extracted line structure caused by the unstable quality of point cloud dataset. Besides, using different persistence threshold $\delta$, the related line structure is extracted from the *LCS'* based on the discrete Morse theory. Moreover, the result is compared with the ground truth line structure and the 3D point cloud segmentation-based line extraction method. In Figure 7c–e, persistence thresholds of 0.01, 6, and 15 are applied to each result, respectively. Although there are more details in Figure 7c, the result seems not good as there are many over-connections in the line structure. However, it does not mean that "the lower the threshold $\delta$, the better the result." In Figure 7e, there can be observed many misconnected line segments with a high threshold $\delta$. Fortunately, the threshold $\delta$ can be interactively set through a trivial operation, and it performs best in the result shown in Figure 7d with threshold $\delta = 6$. Among the results, the line structure extracted based on the proposed method performs superior to others.

The statistic of extracted line structure is conducted, as depicted in Table 2. Line extraction results in Figure 7 are quantitatively compared with the ground truth line structure, and only the line segment within the buffer area (0.05 m) of ground truth line structure is considered to be the effective line structure. The total length of the extracted and effective line structure is calculated. The length of line structure extracted by each method, ranging from Figure 7b to Figure 7e, are 186.94 m, 30.47 m, 24.49 m, and 24.46 m, respectively, while effective lengths is 33.50 m, 23.79 m, 21.39 m, and 21.36 m. Although it has a longer effective length in Figure 7b than the others, there are too many mistakes in the result. In addition, compared with the result in Figure 7d, there are some over-connected line segments in Figure 7c and some misconnected line segments in Figure 7e. Hence, the result of proposed method holds the best performance, and 87.33% of line segments have been effectively extracted from the result. What's more, the connection relation of line structure is preserved, based on the topologic feature of the proposed method.

**Table 2.** The statistic of extracted line structure.

| Different Results | Total Length (m) | Effective Length (m) |
|---|---|---|
| Figure 7a | 17.16 | 17.16 |
| Figure 7b | 186.94 | 33.50 |
| Figure 7c | 30.47 | 23.79 |
| Figure 7d | 24.49 | 21.39 |
| Figure 7e | 24.46 | 21.36 |

*4.5. Line Structure Extraction in the Complex Area*

To make a further assessment of the capability for the proposed method, a complex area of the indoor stair scene is collected by the iPhone-based LiDAR sensor, and experiments are conducted. The stair dataset is in an area of 10.61 m$^2$. To deal with the irregularly distributed sampling density, the dataset is randomly resampled with minimum distance of 0.01 m, and the number of point in the dataset is 141,837, with the average number of 2400 points per m$^2$. As seen from Figure 8a, there are at least 29 facets in different directions. In addition, some structures are incomplete, and the point density is unevenly distributed, which makes it difficult to extract line structures from the quality-unstable dataset.

With neighborhood searching distance $r = 0.1$ m, the initial tensor voting process is conducted, followed by the tensor refinement. Then, the geometric feature of each point is computed based on the refined tensor. In Figure 8b–d, the saliency of surface, line, and point of each point is labeled with the color ranging from light to dark. In Figure 8b, the darker color with the low saliency of "surface" is considered to be the line structure, while the darker color with high saliency of "line" and "point" in Figure 8c,d shows a high probability of being the line structure. Hence, the *LCS* is computed using threshold values $\left\{\sigma_{surface} = 18, \sigma_{line} = 14, \sigma_{point} = 27\right\}$, as depicted in Table 3.

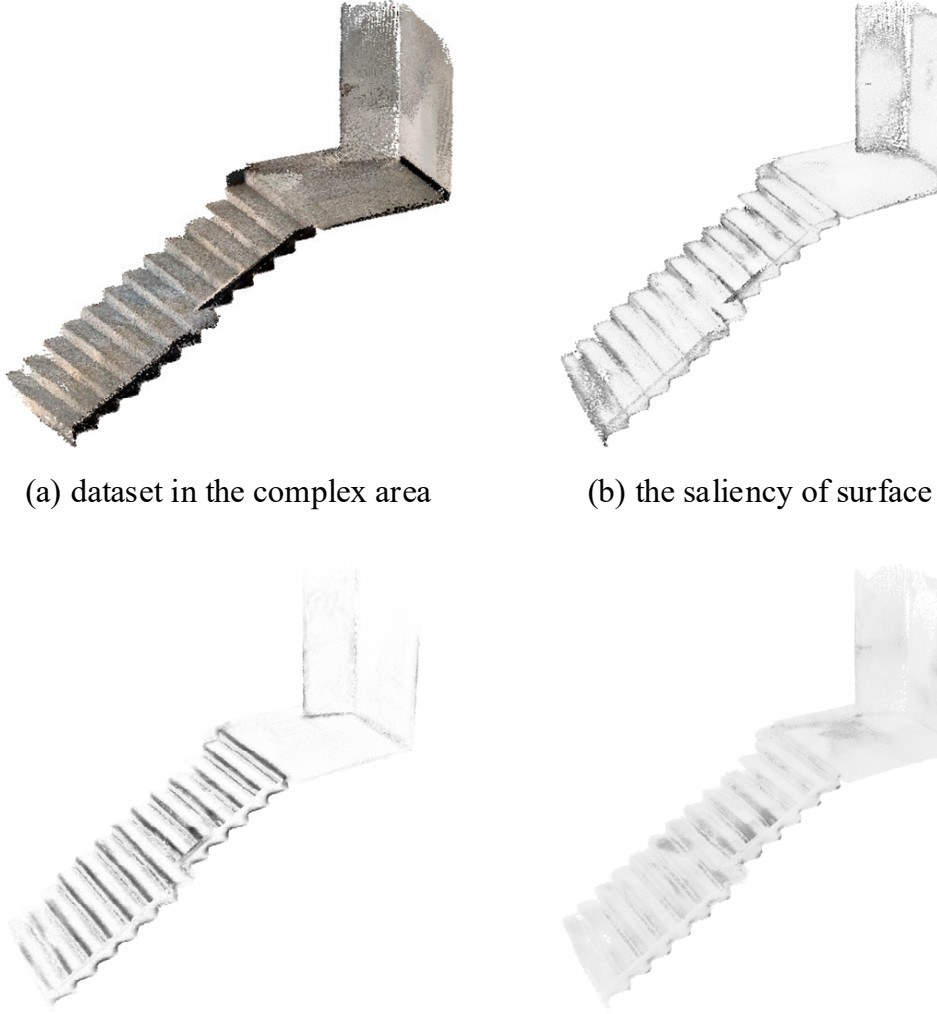

(a) dataset in the complex area

(b) the saliency of surface

(c) the saliency of line

(d) the saliency of point

**Figure 8.** The point cloud in the complex area and the geometric saliency in different dimensions.

**Table 3.** The statistic of *LCS* by different geometric features in the complex area.

| Different Geometric Feature | The Number of Points |
| :---: | :---: |
| *LCS* by $s_{surface}$ | 1225 |
| *LCS* by $s_{line}$ | 20,137 |
| *LCS* by $s_{point}$ | 220 |
| *LCS* without duplication | 21,106 |
| ground truth *LCS* | 48,307 |

As seen in Table 3, the point with surface saliency $s_{surface}$ lower than $\sigma_{surface}$, or line saliency $s_{line}$ higher than $\sigma_{line}$, or point saliency $s_{point}$ higher than $\sigma_{point}$, is selected and taken into the *LCS*. Besides, the number of selected points by $\sigma_{surface}$, $\sigma_{line}$, and $\sigma_{point}$ and is 1225, 20,137, and 220, respectively, and the total point number without duplication is 21,106. On the other hand, the ground truth *LCS* is computed using the spatial relation (within the distance of 0.05 m) between the ground truth line structure and the point cloud, and the total number of points is 48,307. To deal with the quality-unstable point cloud, the *LCS* is resampled, and the related *LCS'* is constructed with different grid sizes $\kappa$, as depicted in Figure 9.

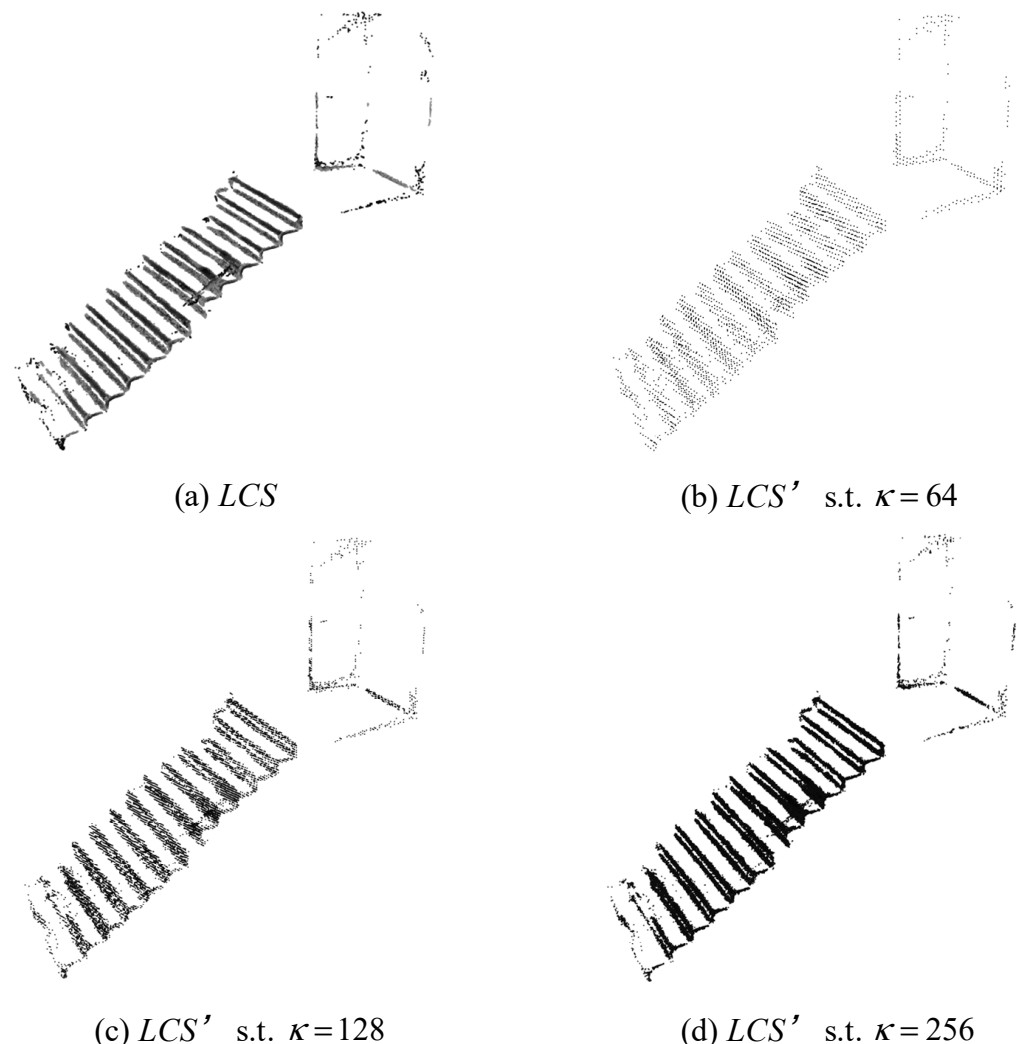

(a) *LCS*  (b) *LCS'* s.t. $\kappa = 64$

(c) *LCS'* s.t. $\kappa = 128$  (d) *LCS'* s.t. $\kappa = 256$

**Figure 9.** The *LCS* and *LCS'* with different grid sizes $\kappa$ in the complex area.

In Figure 9, the *LCS'* is computed with different grid sizes $\kappa$, ranging from 50 to 300. In Figure 9a, it's the constructed *LCS* based on Equation (10). In Figure 9b, with $\kappa = 64$, the details are not properly preserved. While in Figure 9d, the connectivity of line structure can be broken with the high grid size $\kappa = 256$, since the space can be over-divided. To make a balance between the details and the connectivity of line structure, the grid size $\kappa$ is set as 128. Then, the line structure is extracted from the *LCS'* based on the discrete Morse theory, using different persistence thresholds $\delta$, and the result is compared with the ground truth line structure, along with the 3D point cloud segmentation-based line extraction method, as depicted in Figure 10.

Figure 10a shows the ground truth line structure, and it is manually drawn from the original dataset. Figure 10b shows the result computed by the point cloud segmentation-based method [23] (with parameter {Max angle: 10°, Max relative distance: 2.0 m, Max distance: 0.1 m, Min points per facet: 10, Max edge length 0.07 m}), and the line structure is extracted using the rim of segmented geometric objects. In Figure 10c–e, line segments $\{p_{critical}, l_{connection}\}$ are extracted from the *LCS'* with different persistence threshold values $\delta$, ranging from 0.01 to 10, and related line structures $\{\iota\}$ are computed. A low or high threshold $\delta$ will cause the overconnection or misconnection problems in the extracted line structure, as depicted in Figure 10c–e. Here, the optimal threshold $\delta = 0.1$ is interactively set through a trivial operation, as depicted in Figure 10d.

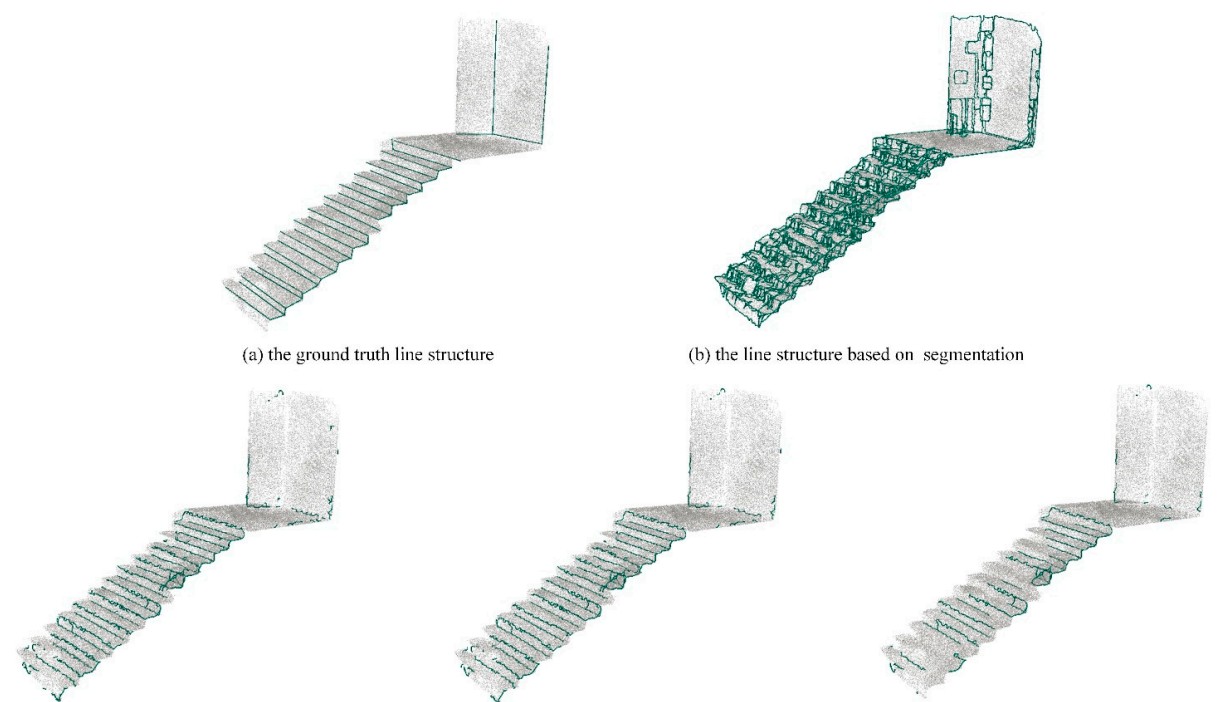

(a) the ground truth line structure

(b) the line structure based on segmentation

(c) the line structure by the proposed method s.t. $\delta = 0.01$ (d) the line structure by the proposed method s.t. $\delta = 0.1$ (e) the line structure by the proposed method s.t. $\delta = 10$

**Figure 10.** The comparison of different line structure extraction results in the complex area.

The related statistics of extracted line structure in the complex area are depicted in Table 4. The total length and effective length of extracted line structure are compared with the ground truth line structure, and only the line segment within buffer area (0.05 m) of the ground truth line segment is considered to be the effective line structure. The length of the ground truth line structure in Figure 10a is 53.98 m, and the total length of line extraction results in Figure 10b–e is 289.58 m, 67.92 m, 67.36 m, and 58.28 m, respectively. However, the effective length of each result is 156.26 m, 56.99 m, 57.77 m, and 43.80 m. Although it has a longer length than others in Figure 10b, there are too many misconnections. With a low threshold value $\delta$, there can be over-connections in Figure 10c, while there can be misconnections in Figure 10e using a high threshold value $\delta$. Among conducted experiments, the line structure extracted by the proposed method (in Figure 10d) outperforms the others. Hence, with 85.76% of line segments effectively extracted, the experiment result in Figure 10d holds the best performance, and it turns out that the proposed method can extract the line structure from the dataset with proper geometric and topological features. In addition, the rate of effectively extracted line segments among experimented datasets remains steady (87.33% for the scene of parterre, 85.76% for the scene of indoor stair), and it turns out to be an effective method to extract line structures in different scenarios.

**Table 4.** The statistic of extracted line structure in the complex area.

| Different Results | Total Length (m) | Effective Length (m) |
|---|---|---|
| Figure 10a | 53.98 | 53.98 |
| Figure 10b | 289.58 | 156.26 |
| Figure 10c | 67.92 | 56.99 |
| Figure 10d | 67.36 | 57.77 |
| Figure 10e | 58.28 | 43.80 |

## 5. Conclusions

With the widespread applications of LiDAR sensors in consumer electronic devices, point cloud datasets in different scenarios are being collected and numerous studies con-

ducted to interpret the 3D scenes and obtain effective representation. Among these approaches is line structure extraction. To deal with the quality-unstable point cloud datasets acquired by consumer electronic devices, the line structure extraction method based on the persistence of tensor feature is applied. The point is encoded and voted to its neighborhood, different geometric features ranging from one dimension to three dimensions are extracted from the voted tensor, and the *LCS* is constructed based on the combination of various features. Then, the Morse–Smale complex is constructed according to the tensor feature, and line structure is extracted using the persistent homology theory. With the point cloud dataset collected by the iPhone-based LiDAR sensor, experiments are conducted, line structure is successfully extracted from the dataset with connection information preserved, and results are compared with related methods. Moreover, using the proposed method, line structure can be extracted from the quality-unstable point cloud dataset, with no predefined geometric models and no manually selected training datasets.

Future research should aim for the refinement of line structure extraction results, since lines are extracted based on the persistence of tensor feature, and they are not purely straight compared with the ground truth dataset. Another aspect that needs to be further explored is improving the robustness of the line structure extraction framework and transforming it into an end-to-end line structure extraction model.

**Author Contributions:** X.W. performed the theory analysis, methodology, performed the experiments, and contributed to drafting the manuscript. H.L. collected and analyzed the data, design, and coding. W.H. performed the literature reviews, improved the writing. Q.C. revised the paper, provided the background knowledge and funding. All authors have read and agreed to the published version of the manuscript.

**Funding:** This research was funded by the National Natural Science Foundation of China (NSFC) (No. 61875088, No. 62005128).

**Institutional Review Board Statement:** Not applicable.

**Informed Consent Statement:** Not applicable.

**Data Availability Statement:** Publicly available datasets were analyzed in this study. The data can be found here: https://doi.org/10.6084/m9.figshare.20407797 (accessed on 20 August 2022).

**Conflicts of Interest:** The authors declare no conflict of interest.

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
