# Peer review of "Line Structure Extraction from LiDAR Point Cloud Based on the Persistence of Tensor Feature"

_applsci, doi:10.3390/app12189190_

Round 1
Reviewer 1 Report
Never introduce a new term without also included its definition. So I read multiple instances of, "line feature," starting in the abstract, but the term is not defined until I read Sec. 2 and even then it isn't clearly defined. This must be fixed.
Why is there a separate sec. 2 and not just one introduction that includes citations to previous related works?
I'd like to see actual source code, perhaps Matlab code, for algorithm 1. It would help greatly if the pseudocode was indented. But really, I'd like to see detailed source code for my review, not necessarily for the published paper.
Way too many "the,"s. It makes it hard to differentiate between a single, accepted method or algorithm versus large classes of algorithms. To be the generally accepted single source algorithm is different from a class of algorithms. Note how I used, "the," for the single accepted versus "a," class of algorithms. Specific example from the text reads, "Another heated research field is the line structure extraction from the point cloud based on the line feature representation." So is "the" line structure extraction a single specific concept or is it a generate notion of extracting straight lines from 3d point clouds? Is "the" point cloud a specific data set that makes all other point clouds irrelevant or do you really want a as is any point cloud dataset?
Author Response
Never introduce a new term without also included its definition. So I read multiple instances of, "line feature," starting in the abstract, but the term is not defined until I read Sec. 2 and even then it isn't clearly defined. This must be fixed.
Response: Accepted. The “line feature”, along with the related description of “line segment”, and “line structure”, has been added to Section 1 and Section 3.
Why is there a separate sec. 2 and not just one introduction that includes citations to previous related works?
Response: It’s just the writing style that we referenced in previously published papers in Applied Science.
I'd like to see actual source code, perhaps Matlab code, for algorithm 1. It would help greatly if the pseudocode was indented. But really, I'd like to see detailed source code for my review, not necessarily for the published paper.
Response: Accepted. Please download the attachment (the key source code) via http://180.209.98.94:8060/s/4QqTdoKJ47iJPmD (password: CXbCPgsH). Since it’s a part of the undergoing project that we have not finished yet, we wouldn’t like it to be open-sourced, while any communication with the purpose of scientific research is welcomed.
Way too many "the,"s. It makes it hard to differentiate between a single, accepted method or algorithm versus large classes of algorithms. To be the generally accepted single source algorithm is different from a class of algorithms. Note how I used, "the," for the single accepted versus "a," class of algorithms. Specific example from the text reads, "Another heated research field is the line structure extraction from the point cloud based on the line feature representation." So is "the" line structure extraction a single specific concept or is it a generate notion of extracting straight lines from 3d point clouds? Is "the" point cloud a specific data set that makes all other point clouds irrelevant or do you really want a as is any point cloud dataset?
Response: Accepted. Related descriptions have been revised. Thanks so much for your valuable comments, and the manuscript has been further improved in this revision.
Reviewer 2 Report
The article is good, but there are some parts that must be improved in the sections:
- Methodology
- Experiments and Discussions
- Conclusions
Line 317: The point cloud dataset is acquired by the LiDAR sensor assembled in the iPhone 12 Pro MaX - it would be recommended to detail the sensor capabilities, and accuracy.
Line 332: the searching distance for the neighborhood is set to be 0.3m – it would be recommended to specify also the point density, as points per square meters, or square centimeters/decimeters, so that readers could make an impression about how many points are located in a range of 0.3 meters.
Line 374: suggested ranging domain is [10,100] as shown on figure 6, for k=15 it seems like the point cloud is very sparse, not reflecting at all the reality, and at r=75 it is a bit closer to the real shape, maybe reconsidering to suggest higher values would be a good.
Line 390: Figure 7 The comparison of different line structure extraction result (a) the ground truth line structure - It would be suggestive if you detail how did you obtained those lines, maybe adding some alternative measurement methods, like a total station survey would add value to the article. Also if you obtained the lines by manually drawing them it would be suggested to specify this.
Line 409: The statistic of extracted line structure. – It would be suggestive besides presenting the lines total length also to compare the location (X, Y, Z) of the shape, and also the shape of the object. Compared as length you may get a lot of noisy lines that are not necessarily representative to the modeled object. Maybe a shape comparison in percent would be more effective, or modeled object volume comparison as percentage.
Line 441: Table 3. The statistic of LCS by different geometric features in the complex area. – it would be more suggestive for the case study to specify besides the number of points also the point density, as points per square meters, or square centimeters/decimeters etc.
Line 464: Figure 10 The comparison of different line structure extraction results in the complex area. (a) the ground truth line structure)- It would be suggestive if you detail how did you obtained those lines, maybe adding some alternative measurement methods, like a total station survey would add value to the article. Also if you obtained the lines by manually drawing them it would be suggested to specify this.
Line 476: Table 4. The statistic of extracted line structure in the complex area – As suggested before, besides comparing the line length it would add value to the article to compare also the location (X, Y, Z) of the shape, and also the shape of the object. A shape modelling fidelity in percentage would be also more suggestive.
Line 493: massive point cloud datasets are collected – As presented on the producer`s page, the IPhone has some limitative capabilities, not quite capable of generating massive point clouds
Also there are no conclusions regarding the compare of line feature extraction from the two datasets, how does the dataset and the scanned object complexity influence the adopted method`s results?
Author Response
The article is good, but there are some parts that must be improved in the sections:
- Methodology
- Experiments and Discussions
- Conclusions
Response: Accepted. Thanks so much for your valuable comments, and the manuscript has been further improved in this revision.
Line 317: The point cloud dataset is acquired by the LiDAR sensor assembled in the iPhone 12 Pro MaX - it would be recommended to detail the sensor capabilities, and accuracy.
Response: Accepted. The related description has been updated.
Line 332: the searching distance for the neighborhood is set to be 0.3m – it would be recommended to specify also the point density, as points per square meters, or square centimeters/decimeters, so that readers could make an impression about how many points are located in a range of 0.3 meters.
Response: Accepted. The related description has been updated.
Line 374: suggested ranging domain is [10,100] as shown on figure 6, for k=15 it seems like the point cloud is very sparse, not reflecting at all the reality, and at r=75 it is a bit closer to the real shape, maybe reconsidering to suggest higher values would be a good.
Response: Accepted. The related description has been updated.
Line 390: Figure 7 The comparison of different line structure extraction result (a) the ground truth line structure - It would be suggestive if you detail how did you obtained those lines, maybe adding some alternative measurement methods, like a total station survey would add value to the article. Also if you obtained the lines by manually drawing them it would be suggested to specify this.
Response: Accepted. The related description has been updated.
Line 409: The statistic of extracted line structure. – It would be suggestive besides presenting the lines total length also to compare the location (X, Y, Z) of the shape, and also the shape of the object. Compared as length you may get a lot of noisy lines that are not necessarily representative to the modeled object. Maybe a shape comparison in percent would be more effective, or modeled object volume comparison as percentage.
Response: It’s a good suggestion. Actually, we have conducted experiments of calculating the overlapped volume between the 3D buffer of each computed line structure and the ground truth line structure, to compare the location and shape of the object. We used software like 3DS MAX, ArcGIS pro, but the software kept crashing when dealing with the process. It still remains a challenging problem to compute the 3D volume of irregular objects, however, we will keep working on it.
Line 441: Table 3. The statistic of LCS by different geometric features in the complex area. – it would be more suggestive for the case study to specify besides the number of points also the point density, as points per square meters, or square centimeters/decimeters etc.
Response: Accepted. The number of points, the minimum resample distance, the point density (points per square meters) of two experimented datasets have been provided, and the related description has been updated. hence, the point density of LCS isn’t needed, since it’s the same with the original dataset.
Line 464: Figure 10 The comparison of different line structure extraction results in the complex area. (a) the ground truth line structure)- It would be suggestive if you detail how did you obtained those lines, maybe adding some alternative measurement methods, like a total station survey would add value to the article. Also if you obtained the lines by manually drawing them it would be suggested to specify this.
Response: Accepted. The related description has been updated.
Line 476: Table 4. The statistic of extracted line structure in the complex area – As suggested before, besides comparing the line length it would add value to the article to compare also the location (X, Y, Z) of the shape, and also the shape of the object. A shape modelling fidelity in percentage would be also more suggestive.
Response: It’s a good suggestion. Actually, we have conducted experiments of calculating the overlapped volume between the 3D buffer of each computed line structure and the ground truth line structure, to compare the location and shape of the object. We used software like 3DS MAX, ArcGIS pro, but the software kept crashing when dealing with the process. It still remains a challenging problem to compute the 3D volume of irregular objects, however, we will keep working on it.
Line 493: massive point cloud datasets are collected – As presented on the producer`s page, the IPhone has some limitative capabilities, not quite capable of generating massive point clouds
Response: Accepted. The related description has been updated.
Also there are no conclusions regarding the compare of line feature extraction from the two datasets, how does the dataset and the scanned object complexity influence the adopted method`s results?
Response: Accepted. The related description has been updated.
Round 2
Reviewer 2 Report
The article is much improved.